# Il progetto OpenAcolit, un repertorio delle biblioteche italiane realizzato con Wikibase

## Autore

Stefano Bargioni, ORCID 0000-0001-7679-2874

## Abstract

OpenAcolit, successore di Acolit, è un progetto innovativo sviluppato da ABEI, PUSC, BCE-CEI e DMBC-UniPV. Non si propone di trascrivere i volumi cartacei del repertorio Acolit, ma di portarne i dati nel web semantico tramite Wikibase Cloud. Questa opera digitale è impostata sulla collaborazione tra bibliotecari e studiosi, offre nuove modalità di fruizione e favorisce l'interazione con altri software.

OpenAcolit, the successor to Acolit, is an innovative project developed by ABEI, PUSC, BCE-CEI, and DMBC-UniPV. It does not aim to transcribe the paper volumes of the Acolit repertoire but rather to bring its data into the semantic web through Wikibase Cloud. This digital initiative is built on collaboration between librarians and scholars, offering new ways of access and interaction with other software.

## Keywords

OpenAcolit, Acolit, Wikibase, authority list, linked open data.

## Introduzione

OpenAcolit[1] succede ad Acolit, "Autori cattolici ed opere liturgiche", opera in 4 volumi pubblicata dal 1998 al 2010 da ABEI. OpenAcolit è un progetto di ABEI, PUSC, BCE-CEI, DMBC-UniPV.[2]

---

[1] https://openacolit.wikibase.cloud. Questo link e tutti i link citati di seguito sono stati verificati ad aprile 2025.

[2] ABEI - Associazione dei Bibliotecari Ecclesiastici Italiani; PUSC - Pontificia Università della Santa Croce; BCE-CEI - Ufficio Nazionale per i beni culturali ecclesiastici e l'edilizia di culto della Conferenza Episcopale Italiana; DMBC-UniPV - Dipartimento di Musicologia e Beni Culturali dell'Università degli Studi di Pavia.

La natura di Acolit è quella del repertorio, della risorsa di riferimento per la corretta dicitura di nomi di autori, enti, opere legati ad uno dei più ampi ambiti della nostra cultura, quello ecclesiologico, dove storia, diritto, teologia e filosofia si richiamano costantemente.[3] OpenAcolit intende aggiornare Acolit sia in quanto al contenuto, sia in quanto alle funzionalità. Forse gli anni trascorsi dall'inizio della sua stesura non danno l'impressione di grandi cambiamenti nelle corrette forme di citazione delle entità suddette. Tuttavia la sola ipotesi di procedere ad una nuova edizione cartacea per introdurvi gli aggiornamenti ha costituito per diversi anni un blocco psicologico difficilmente sormontabile per ABEI. Anche l'ipotesi di concentrare questo compito su poche persone, pur passando a una base informatica, non permetteva di prendere una decisione.

La soluzione proposta, più avanti illustrata in maggior dettaglio, intende basarsi sul contributo di un gruppo aperto di bibliotecari direttamente interessati all'utilizzo dei dati, di esperti e studiosi, così come intende contribuire al Controllo Bibliografico Universale (UBC).[4] Wikibase[5], e più precisamente in Wikibase Cloud, si è dimostrato in grado di adempiere ai due compiti fondamentali suddetti.[6] Una volta creata l'istanza OpenAcolit in wikibase.cloud, su di essa sono stati intrapresi lo studio di fattibilità, i primi test reali e le prime pratiche di scrittura concorrenziale dei dati.

Il progetto è stato presentato ed utilizzato in tre diverse giornate di incontri di bibliotecari e bibliotecarie, tenutesi in autunno 2024 a Roma, Andria e Milano, grazie anche alla fattiva cooperazione di Wikimedia Italia.

# Significati di "open" in OpenAcolit

L'aggiunta della parola "open" al nome di Acolit è stato un passo quasi naturale, ma nel tempo si è dimostrata essere una chiave di lettura molto trasversale del progetto:

- chiunque lo può consultare e ne può utilizzare i dati, compresi gli URI
- chiunque, con la sufficiente formazione professionale,[7] può concorrere al mantenimento
- non ha carattere commerciale, implicito nella pubblicazione di un'opera cartacea o di un database ad accesso controllato con credenziali
- va oltre il contenuto di Acolit, permettendo aggiunte e correzioni
- va potenzialmente oltre le tipologie di entità trattate nei 4 volumi, per esempio potrebbe includere i santi, risolvendo la problematica della duplicazione causata da papi santi

---

[3] Per il posizionamento di Acolit nel lavoro di catalogazione delle biblioteche italiane e il suo rapporto con REICAT e Nuovo Soggettario, si veda Guerrini (2009).

[4] Per il progressivo sviluppo dell'UBC si può vedere Sardo (2021).

[5] Wikibase è un software libero e open source realizzato e supportato da Wikimedia Deutschland e da una vasta comunità di collaboratori. Wikibase è disponibile in cloud (https://www.wikibase.cloud) o installabile in locale (https://wikiba.se/). In entrambi i casi è gratuito. Anche Wikidata è realizzato con Wikibase.

[6] Il rapporto tra UBC e Wikidata, e per analogia anche tra UBC e sistemi basati su Wikibase, è esaminato in Bianchini, Sardo (2022).

[7] Per diventare utenti e poter editare OpenAcolit, occorre richiedere l'accesso. Non sono ammesse la creazione automatica di un account né le modifiche anonime, come invece accade in Wikidata o in Wikipedia.

- oltre alla forma di Acolit, riporta le forme costruite in base a normative catalografiche di ambiti diversi, vigenti, in disuso o di prossima introduzione
- può andare oltre l'ambito cattolico e includere altre confessioni
- può andare oltre l'ambito della lingua italiana, anche nella lingua dell'interfaccia
- è potenziale strumento di studio comparativo ed evolutivo di normative
- è potenziale strumento di migrazione delle forme usate in un catalogo da una normativa all'altra
- è attore sul palco dei LOD: genera URI per ogni entità, è unito a Wikidata tramite apposita proprietà ed è consultabile con API e query SPARQL, anche federate.[8]

# Stato del progetto

Acolit raccoglie le forme di nomi relativi a:
- Bibbia, Chiesa cattolica, Curia romana, Stato pontificio, Vaticano, papi e antipapi
- Ordini religiosi
- Opere liturgiche
- Padri della Chiesa e scrittori ecclesiastici occidentali (secoli XII-XIII).

La duttilità di un database di triple RDF su cui si basa OpenAcolit, consente di definire proprietà a sostegno di tipi diversi di entità. Quella fondamentale è "istanza di", che permette di distinguere i vari tipi o classi. Anche la proprietà che unisce un'istanza di OpenAcolit al corrispondente elemento di Wikidata è comune a tutte le classi. Ogni classe ha poi proprietà specifiche. La query seguente

```
PREFIX oawdt: <https://openacolit.wikibase.cloud/prop/direct/>
PREFIX wikibase: <http://wikiba.se/ontology#>
SELECT ?istanzaLabel (COUNT(?q) AS ?count) WHERE {
  ?q oawdt:P1 ?istanza.
  SERVICE wikibase:label { bd:serviceParam wikibase:language
"[AUTO_LANGUAGE],en". }
}
GROUP BY ?istanza ?istanzaLabel
```

| istanzaLabel | count |
|---|---:|
| res | 2 |
| opera | 118 |
| espressione | 931 |
| lingua | 165 |
| tipo di fonte | 6 |

---

[8] Le query federate sono un tipo di interrogazione di database RDF in cui possono essere richiesti dati contemporaneamente a più di un server SPARQL. SPARQL è acronimo ricorsivo di "SPARQL Protocol and RDF Query Language", ed è un linguaggio di interrogazione standard per il recupero e la manipolazione dei dati memorizzati in formato RDF.

mostra le istanze attualmente definite in OpenAcolit.[9] Come si nota, sono relative a opere, e in particolare a opere della Bibbia e a loro espressioni in lingue diverse. Come forse è stato già intuito, lo scopo biblioteconomico e quindi non letterario, né storico o altro, ha portato a limitare la descrizione delle opere e delle espressioni raccogliendo per ognuna la forma prevista da una o più normative catalografiche o fonti.

Ad oggi le fonti sono 6: ACOLIT, REICAT, AACR2, URBE, BAV, Volpi.[10] Dal lato pratico, quindi, un'agenzia di catalogazione cercherà tra le forme della normativa di proprio riferimento. Probabilmente OpenAcolit verrà adoperata principalmente per le REICAT, ma potrà fare lo stesso esatto servizio per altre normative o "varianti locali" che nel tempo verranno introdotte in OpenAcolit.

Sarà stato notato l'utilizzo del modello IFLA-LRM.[11] Ogni espressione, come può essere un libro dell'Antico o del Nuovo Testamento in una determinata lingua, realizza l'opera corrispondente, e il legame tra le due è rappresentato da una apposita proprietà,[12] nel verso di "molte a una".

L'esame di un elemento di OpenAcolit, l'opera "Bibbia. A.T. Tobia"[13] consente di comprendere meglio diversi aspetti e potenzialità di OpenAcolit.

```
istanza di          opera
lingua              ebraico
wikidata            Q131737[14]
è parte di          Bibbia
                    Bibbia. A.T.
                    Bibbia. A.T. Scritti
                    Bibbia. A.T. Libri storici
                    Bibbia. A.T. Deuterocanonici
forma autorevole    Bibbia. A.T. Tobia
                        fonte ACOLIT
                    Bibbia. Antico Testamento. Tobia
                        fonte REICAT
                    Biblia. V.T. Tobias
                        fonte BAV
                    Biblia. V.T. Thobias
                        fonte Varianti Locali URBE
```

Ogni occorrenza della proprietà "forma autorevole" (P27) è distinta dalle altre dal qualificatore "fonte". Questo raggruppamento di forme, ognuna autorevole nell'ambito di applicazione identificato dalla fonte stessa, è al momento -per quanto è dato sapere- un *unicum* nel mondo della biblioteconomia. Come accennato in precedenza, P27 è la base per il confronto tra fonti, anche tra fonti ormai non più vigenti ma eventualmente ancora presenti

---

[9] I risultati sono aggiornati al 2.4.2025.

[10] La fonte Volpi si riferisce al repertorio "Vittorio Volpi, "DOC : dizionario delle opere classiche : intestazioni uniformi degli autori, elenco delle opere e delle parti componenti, indici degli autori, dei titoli e delle parole chiave della letteratura classica, medievale e bizantina", Milano : Editrice bibliografica, 1994", ed è stata introdotta in OpenAcolit non per riportare i dati del repertorio stesso, ma per poterlo usare come citazione in occasioni particolari.

[11] https://www.ifla.org/wp-content/uploads/2019/05/assets/cataloguing/frbr-lrm/ifla-lrm-august-2017_rev201712-it.pdf.

[12] https://openacolit.wikibase.cloud/wiki/Property:P7.

[13] L'URI è https://openacolit.wikibase.cloud/entity/Q67.

[14] A sua volta, e a suo tempo, Wikidata potrà contenere Q67, e in generale gli identificatori di OpenAcolit, in un'apposita proprietà.

in cataloghi non convertiti. Va notata anche la possibilità di sviluppare una propria nuova normativa, o di trascrivere in supporto informatico una normativa vigente ma regolamentata ancora su supporto cartaceo, permettendo così che interagisca con gli attuali sistemi di catalogazione e metadatazione.[15]

Per gli altri tipi di entità trattati da Acolit, OpenAcolit si svilupperà in modo analogo come fatto con opere ed espressioni della Bibbia: definizione delle proprietà specifiche, conversione dei dati dai file di Acolit, loro importazione automatica, controllo manuale della completezza e della qualità dell'importazione, completamento ed aggiornamento, anche a consultazione in corso.[16]

# Esempi di utilizzo

## In catalogazione

L'utilizzo più frequente che si può ipotizzare per OpenAcolit è costituito dal recupero della corretta forma da impiegare in catalogazione. Nel caso di opere ed espressioni della Bibbia e in ambito MARC21, i tag interessati sono 130, 730, 630, che in ambito Unimarc (SBN) corrispondono ai tag 740, 742, 606.

È possibile ricercare la forma in OpenAcolit, copiarla e riportarla nel proprio catalogo; ma è anche possibile, e molto più produttivo, associare una funzionalità di autocompletamento (o autosuggest) basata sull'interrogazione automatica dell'endpoint SPARQL di OpenAcolit.[17]

La variabile "query" impostata con il testo del codice SPARQL seguente

```
#title: Autosuggest per "apocalisse" (REICAT)
PREFIX rdfs: <http://www.w3.org/2000/01/rdf-schema#>
PREFIX oap: <https://openacolit.wikibase.cloud/prop/>
PREFIX oaps: <https://openacolit.wikibase.cloud/prop/statement/>
PREFIX oapq: <https://openacolit.wikibase.cloud/prop/qualifier/>
PREFIX oawd: <https://openacolit.wikibase.cloud/entity/>
PREFIX oawdt: <https://openacolit.wikibase.cloud/prop/direct/>
PREFIX wikibase: <http://wikiba.se/ontology#>
PREFIX bd: <http://www.bigdata.com/rdf#>
SELECT ?q ?forma ?fonteLabel WHERE {
  ?q rdfs:label ?l;
     oawdt:P1 ?istanza;
     oawdt:P17 ?forma;
     oap:P17 _:b26.
  _:b26 oaps:P17 ?forma;
     oapq:P16 ?fonte.
  values ?fonte {oawd:Q1236} # Q1236 REICAT ; Q1243 URBE ; Q1242 AACR2
  FILTER((LANG(?l)) = "it")
  FILTER(CONTAINS(LCASE(?l), "apocalisse"))
```

---

[15] Potrebbe essere il caso delle "Varianti locali" di URBE, o della Biblioteca apostolica vaticana, per la quale sono vigenti in cartaceo le "Norme per il catalogo degli stampati", terza ed., 1949.

[16] Il Gruppo Promotore di OpenAcolit non ha ancora definito successione e tempistiche per il trattamento dei dati rimanenti di Acolit.

[17] L'endpoint SPARQL di OpenAcolit si trova all'indirizzo https://openacolit.wikibase.cloud/query.

```
        SERVICE wikibase:label { bd:serviceParam wikibase:language "it". }
    }
    ORDER BY asc(?forma)
```

può essere adoperata con successo all'interno di un ambiente di catalogazione che
interroghi OpenAcolit, per esempio con il seguente codice JavaScript/jQuery:

```
    let Q = encodeURIComponent(query);
    jQuery.getJSON(`https://openacolit.wikibase.cloud/query/sparql?query=
    ${Q}`, function(R){
        // utilizzo della risposta R
    })
```

In questo caso, insieme alla forma si ha anche il vantaggio della registrazione automatica
dell'URI di OpenAcolit, da conservare in un apposito sottocampo del tag da popolare.[18]
È decisivo riportare l'URI dell'entità nei propri dati allo scopo di legare questi ultimi ai Linked
Open Data a cui OpenAcolit appartiene per costituzione. E nel caso di forma mancante,
registrarla in OpenAcolit anche subito prima di procedere al suo utilizzo, permette non solo
di fissare la stringa autorevole (almeno quella della normativa di proprio interesse) e possibili
varianti, ma anche di assegnare un URI all'entità.

## Per studio e ricerca

Un ulteriore caso è rappresentato dallo studio dei dati di OpenAcolit. In questo esempio si
confrontano le forme di alcune espressioni dell'opera Apocalisse e di alcune sue
espressioni.

```
    #title: Confronto tra le forme REICAT e URBE per opere ed
    espressioni dell'Apocalisse
    PREFIX rdfs: <http://www.w3.org/2000/01/rdf-schema#>
    PREFIX oap: <https://openacolit.wikibase.cloud/prop/>
    PREFIX oaps: <https://openacolit.wikibase.cloud/prop/statement/>
    PREFIX oapq: <https://openacolit.wikibase.cloud/prop/qualifier/>
    PREFIX oawd: <https://openacolit.wikibase.cloud/entity/>
    PREFIX oawdt: <https://openacolit.wikibase.cloud/prop/direct/>
    PREFIX wikibase: <http://wikiba.se/ontology#>
    PREFIX bd: <http://www.bigdata.com/rdf#>
    SELECT ?forma ?fonteLabel WHERE {
      ?q rdfs:label ?l;
          oawdt:P1 ?istanza;
          oawdt:P17 ?forma;
          oap:P17 _:b26.
      _:b26 oaps:P17 ?forma;
          oapq:P16 ?fonte.
      VALUES ?fonte {
          oawd:Q1243
```

---

[18] Va anche notato che la risposta è di tipo CORS: tutte le basi Wikibase includono nei dati della
risposta la dichiarazione HTTP "`access-control-allow-origin: *`" che permette a un browser
web di accettare i dati ricevuti da terze parti come riutilizzabili, mentre ciò ordinariamente non
avviene. Per una trattazione esaustiva della problematica, si veda
https://developer.mozilla.org/en-US/docs/Web/HTTP/Guides/CORS/Errors.

```
        oawd:Q1236
    }
    FILTER((LANG(?l)) = "it")
    FILTER(CONTAINS(LCASE(?l), "apocalisse"))
    SERVICE wikibase:label {
 bd:serviceParam wikibase:language "it". }
    }
ORDER BY (?q) (?fonteLabel)
```

La risposta è

```
Bibbia. Nuovo Testamento. Apocalisse                  REICAT
Biblia. N.T. Apocalypsis                              URBE
Bibbia. Nuovo Testamento. Apocalisse (in francese)    REICAT
Biblia. N.T. Apocalypsis. Francese                    URBE
Bibbia. Nuovo Testamento. Apocalisse (in inglese)     REICAT
Biblia. N.T. Apocalypsis. Inglese                     URBE
Bibbia. Nuovo Testamento. Apocalisse (in italiano)    REICAT
Biblia. N.T. Apocalypsis. Italiano                    URBE
Bibbia. Nuovo Testamento. Apocalisse (in latino)      REICAT
Biblia. N.T. Apocalypsis. Latino                      URBE
Bibbia. Nuovo Testamento. Apocalisse (in spagnolo)    REICAT
Biblia. N.T. Apocalypsis. Spagnolo                    URBE
Bibbia. Nuovo Testamento. Apocalisse (in tedesco)     REICAT
Biblia. N.T. Apocalypsis. Tedesco                     URBE
```

# Conclusioni

OpenAcolit possiede diverse caratteristiche della tradizione repertoriale cartacea. Non trattandosi però di una trascrizione di Acolit, è più appropriato dire che si tratta di una trasposizione dei suoi dati. OpenAcolit li rende fruibili in modo completamente diverso, li integra nel web semantico e ne permette il riuso con altri software. Ancor più, la compartecipazione nella sempre aperta redazione di numerosi attori sufficientemente esperti in ogni settore può assicurare la continuità delle authority list di Acolit.
OpenAcolit quindi rappresenta una nuova frontiera del concetto di repertorio. Senza esserselo proposto, risponde in modo nuovo alle domande che pose Barbara Tillett in una intervista a Mauro Guerrini riportata nella presentazione del Volume 2 di Acolit, in particolare alle domande su forme in altre lingue, sulla possibilità e modalità di aggiornamento dell'opera, sullo sfruttamento informatico dei dati, e soprattutto sulla possibilità di riportare le diverse forme distinte per fonte (RICA, AACR2, RAK…).[19] Significativamente, e quasi profeticamente, Guerrini conclude l'ultima risposta con le seguenti parole: "Speriamo nella collaborazione dei bibliotecari e dei lettori che lavorano in questo settore". Speranza che OpenAcolit riaccende e, almeno potenzialmente, può soddisfare.

---

[19] Acolit, Editrice Bibliografica, 2000, Volume 2, pp. IX-XI. La versione inglese è a pp. XXXIII-XXXV.

# Bibliografia

Guerrini (2009), "Il catalogo "ecclesiastico" oggi tra Acolit, nuove RICA e nuovo soggettario", in Bollettino ABEI 2/2009 https://hdl.handle.net/2158/1132972.

Sardo (2021), *L'evoluzione dell'authority control*, in "La trasmissione della conoscenza registrata: scritti in onore di Mauro Guerrini offerti dagli allievi", a cura di Carlo Bianchini e Lucia Sardo", 379-390.

Bianchini, Sardo (2022). "Wikidata: A New Perspective towards Universal Bibliographic Control". JLIS.It 13 (1):291-311. https://doi.org/10.4403/jlis.it-12725

