# OpenReview forum: "Il progetto OpenAcolit, un repertorio delle biblioteche italiane realizzato con Wikibase"
_wikimedia.it/Wikidata_and_Research/2025/Conference — WD&R Paper_

### Official Review · ~Lucia_Sardo1 · 2025-01-04
**Revisione "sulla fiducia"**

**Originality:** 5
**Impact:** 5
**Confidence:** 5

**Review:**

Sebbene il paper non abbia un abstract e si possano dedurre qualità rilevanza e impatto solo dal titolo, è sicuramente una presentazione di grande interesse in particolare per la comunità bibliotecaria, ma non solo. Si tratta infatti della trasposizione in LOD dell'authority list dedicata agli autori cattolici e alle opere liturgiche pubblicata alla fine del secolo scorso. In questo senso un lavoro di questo tipo è di grande impatto e interesse per tutti coloro che a vario titolo potrebbero essere interessati a questo tipo di dati. A mio avviso si tratta di un progetto la cui realizzazione porterà vantaggi e un miglioramento della qualità dei dati su queste entità.

**Compliance:**

5

**Scientific Quality:**

5

---

### Official Review · ~Luca_Martinelli1 · 2025-01-04
**Accettato sulla fiducia**

**Originality:** 4
**Impact:** 5
**Confidence:** 5

**Review:**

Concordo con la revisione fatta da Lucia Sardo. Lo scopo del progetto portato avanti da Utente:Bargioni è sicuramente di alto impatto per la comunità di Wikidata, dal momento che va a coprire un ambito estremamente interessante e particolare da coprire, con dei dati di sicuro valore data la fonte. Sarà sicuramente un progetto interessante da sentire, soprattutto per quanto riguarda lo svolgimento delle attività.

**Compliance:**

5

**Final Paper Review:**

Anche in questo caso, confermo il mio giudizio positivo. Sebbene il paper sia un po' corto e specifichi solo in ultimo i vantaggi del lavoro svolto (forse questi potrebbero essere resi un po' più espliciti, in effetti), si tratta comunque di un paper di sicuro interesse per l'area biblioteconomica, in un ambito come quello dei testi ecclesiastici che è particolarmente delicato.

**Scientific Quality:**

5

---

### Decision · Program_Chairs · 2025-02-05

Accept (Paper)